# Femtosecond Pump Probe Reflectivity Spectra in CdTe and GaAs Crystals at Room Temperature

**DOI:** 10.3390/ma13010242

**Published:** 2020-01-06

**Authors:** Hao Sun, Hong Ma, Jiancai Leng

**Affiliations:** 1Shandong Provincial Key Laboratory of Optics and Photonic Device and Collaborative Innovation Center of Light Manipulations and Applications, School of Physics and Electronics, Shandong Normal University, Jinan 250014, China; sunhaosdnu@stu.sdnu.edu.cn; 2School of Electronic and Information Engineering (Department of Physics), Qilu University of Technology (Shandong Academy of Sciences), Jinan 250353, China

**Keywords:** GaAs and CdTe crystals, pump probe reflectivity spectroscopy, BF and BGR effects

## Abstract

Ultrafast pump probe reflectivity (PPR) signal near band edge is modeled by taking into account band filling (BF) and band gap renormalization (BGR) effects with the carrier density of ~10^17^/cm^3^ in GaAs crystal at room temperature. The calculated results indicate that the transient reflectivity Δ*R*/*R* is determined by BF and BGR effects. The most interesting feature is that Δ*R*/*R* signal experiences a sign change from photo-bleaching (PB) to photo-absorption (PA) due to the competition between BF and BGR effects. We experimentally measured Δ*R* as a function of photon energy across band edge with carrier density of ~10^17^/cm^3^ in GaAs and CdTe crystals, which has a similar trend as that calculated according to our model. In addition, the reflectivity is very sensitive to electron spin orientation, which is well confirmed by the corresponding experiments with 100 fs pump probe reflectivity spectroscopy in bulk CdTe. Our research in this work provides a method to study optoelectronic properties of conventional semiconductors at moderate carrier density excited by ultrafast laser pulse. Importantly, this model can be used for other novel semiconductor materials beyond GaAs and will provide new insights into the underlying spin dependent photophysics properties for new materials.

## 1. Introduction

The study of the nonequilibrium carrier dynamics after excited by ultrafast laser pulse in semiconductors and semiconductor nanostructures has always been a very hot topic owing to its wide applications, such as lasers, photodetectors, and light emitting diodes [1,2,3,4,5,6]. Time resolved pump probe spectroscopy is one of the popular methods, which has proved to be a simple but powerful technique to characterize optical properties of photo-excitation carriers [7,8,9]. As for the thick and nontransparent samples, reflected geometry is much more effective by taking advantage of reflectivity instead of transmittance. There has been numerous time resolved pump probe reflectivity (PPR) measurements reported to study carrier lifetime and dynamics in semiconductors [10,11,12,13]. However, the interpretation of the experimental results sometimes can be daunting even for experienced researchers because the dynamics are susceptible to carrier density, temperature, polarization, etc. In general, the PPR response in semiconductor crystals can be attributed to the effects of photo-excited carriers generated by the pump pulses via interband transitions. The diverse carrier relaxation, decay mechanisms, and absorption after photo-excitation may lead to a change in the complex dielectric function [14,15,16]. These changes can be modeled by considering the change of absorption coefficient (Δα) due to BF effect, BGR effect, intraband absorption by the free carriers (FCA), and so on. These effects are determined by many factors and compete with each other. Especially, under high nonequilibrium carrier distribution and high photo-excited energy above the band gap, the recovery process is accompanied by an initial thermalization and followed by a fast cooling process [17]. Therefore, the carrier relaxation is a complicated process that normally cannot be described by single exponential decay.

During the past several decades, the ultrafast optical response under high carrier density in semiconductors had been studied in both theory and experiment. The change in refractive index ∆n in InP, GaAs, and InGaAsP was theoretically estimated by fully taking the BF effect, BGR effect, and FCA into account [14]. The refractive index change reaches 10^−2^ with the carrier density of 10^18^/cm^3^. The PPR spectra in GaAs and GaN were investigated in Reference 15, which was dependent on delay time and carrier density. They pointed out that the magnitude and signature of the PPR signal at different delay times were governed by interplay between BF and BGR effects. Recently, the change in transmittance and reflectivity were also studied based on BF and BGR effects in perovskites emerging as a promising solar cell material [18,19,20,21,22]. A slow hot carrier cooling process [21] and hot phonon bottleneck effect [20] were studied in lead halide perovskites. The optical properties in organometal halide perovskites were observed and calculated by considering BGR, BF, and many body effects [18,19,22]. However, these research works mentioned above were carried out with linearly polarized laser pulses which provided little insight into the influence of carrier spin on PPR spectra. Spin dependent many body effects were studied by time and polarization resolved transmission measurements in GaAs [23]. The spin dependent reflectivity spectroscopy in InP was observed just only at 70 K [17]. However, the optical properties at room temperature are much more important for practical applications.

Spin dependent reflectivity as well as transmission spectroscopy are extremely important, and they can study optical spin properties, especially for thick non-transparent samples for spintronics. As we all know, semiconductor spintronics aims at utilizing or incorporating the spin degree of freedom in a carrier for a new generation of spintronic devices that are more stable, smaller in size, faster, and have lower power consumption than traditional electronic devices. The spin dependent optical parameters are the basis of designing spintronics devices. Therefore, we calculated the change in reflectivity (Δ*R*/*R*) before and after excited by ultrafast laser pulse considering BF and BGR effects. It is worth mentioning that we studied a spin dependent reflectivity signal $\Delta R^{\pm }/R^{\pm }$ excited by circular polarization laser pulse as a function of photon energy in GaAs. Our model established in this work can also be used in other new direct semiconductors and has insight into spin dependent properties. We correlated the theoretical results with the experiments by utilizing PPR spectroscopy with 100 fs laser. In Section 2, the samples and experimental setup were briefly introduced. In Section 3, we described an appropriate theoretical model that incorporates some relevant effects at moderate density. In Section 4, we used the model to simulate the PPR signal and directly interpreted the observed experimental results.

## 2. Materials and Methods

The samples adopted here were a 450 μm-thickness-CdTe and a 1 mm-thickness-GaAs a single crystal (purchased from Hefei Kejing Materials Technology co., LTD., Hefei, China), which were not intentionally doped. The absorption spectra at room temperature are shown in Figure 1 (black line for GaAs and red line for CdTe). A band edge absorption is clearly observed. The absorbance is nearly zero below the band gap energy, then rapidly rises and reaches saturation at last. The Tauc plots of materials (blue and green lines) are shown in Figure 1. According to the absorption spectra and assuming parabolic direct band edge, the band gap energies of bulk CdTe and GaAs are about 1.51 eV and 1.38 eV, respectively. The inset in Figure 1 illustrates the band structure at Γ point of the first Brillouin zone and the selection rules photo-excited by circularly polarized light. The electron spin polarization is generated by a circularly polarized laser pulse according to the conservation of angular momentum for transitions. In direct semiconductor like GaAs, for photon energies equal or slightly superior to the band-gap energy *E_g_*, the transitions can take place between the top of the degenerate valence band P_3/2_ at Γ_8_ and the bottom of the conduction band S_1/2_ at Γ_6_. The ratio of the optically generated spin-up electrons n↑ from heavy hole (HH) band P_3/2_ to spin-down electrons n↓ from light-hole (LH) band P_1/2_ is 3:1 (excited by left-handed circularly polarized light, $\sigma^{-}$) owing to the form of the matrix elements for HH and LH interband transitions. Therefore, the maximum degree of net spin polarization is 0.5 (P=(n↑−n↓)/(n↑+n↓)=0.5) in bulk semiconductor [13,24]. The transitions between split-off band P_1/2_ (SO) at Γ_7_ and conduction band S_1/2_ at Γ_6_ was forbidden by controlling photon energy smaller than $E_{g}+\Delta$, where Δ is the energy difference of Γ_7_ and Γ_8_.

Optical measurements of carrier dynamics were carried out using the time resolved PPR technique. The pulse from a tunable mode-locked Ti: Sapphire laser (Maitai 1020, 100 fs, 80 MHz) was divided into two beams by a beam splitter. The reflected strong one was used as pump beam, which was chopped at a frequency of 1 kHz by an optical chopper. The transmitted weak beam was used as probe beam and passed through an optical delay line driven by a computer-control-step-motor in order to change the delay time between pump and probe light. The intensity ratio of pump to probe pulses was kept to be more than 10:1. The two beams were focused by a 30 cm-focal-length lens and overlapped on the sample surface with a spot size of 100 μm. The pump beam reflected by the sample was blocked with a non-transparent aperture, while the probe beam was guided into a silicon photo-detector connected with a lock-in amplifier. The photon energy was continuously tuned around the band gap energy, which was impossible to reach the transition between the split-off (SO) band and the conduction band. The spin polarization was controlled by two commercial wide band quarter wave plates. A continuous variable attenuator was used to control pump laser power so that the carrier density was kept the same in all experiments. All experiments were conducted at room temperature.

## 3. Theory Model

The electron excited by ultrafast pump pulse transits to conduction band and leaves a hole in valence band in the form of Fermi–Dirac distribution. The carrier accumulation in the finite levels at a conduction band causes an increase of the intrinsic band gap energy, which has been explained by the BF effect. The broadening of the band gap directly leads to a change in absorption coefficient and corresponding variation in refractive index. On the other hand, the photo-excitation electrons occupy the band edge states at the bottom of conduction band, which causes the electron wave functions to overlaps each other. The electrons with the same spin repel and avoid one another. Therefore, the final result is that the energy of the bottom of conduction band becomes lower. The same also occurs for the holes left in the valence band, which leads to an increase in their energy at the top of the valence band. Therefore, the interaction between electrons and holes induces the band gap shrinking, namely, the BGR effect. Similar to the BF effect, the BGR effect can also arouse a change in absorption coefficient and refractive index. Compared to BF and BGR effects, FCA is too small to consider when the density is lower than 1 × 10^18^ /cm^3^ [15]. As a matter of fact, the BF and BGR effects interplay, compete, and cannot be separated.

In what followed, a quantitative interpretation of the change in optical properties mentioned above was systematically studied. We succinctly described the theoretical model on PPR response by taking BF and BGR effects into account. The detailed theoretical modeling which simulates the PPR signal under various conditions of carrier relaxation with a linear polarization laser was obtained in [15].

In this model, the parabolic bands are adopted in a direct band gap semiconductor and the optical absorption near the band gap excited by a linearly polarized probe light is given:(1)$$\alpha (E)=\sum_{\nu }C_{\nu }\sqrt{E-E_{g}}/E\;\;E\geq E_{g},$$
where *E_g_* is the band gap energy, E=hν is the photon energy, ν = *HH*, *LH*, and $C_{\upsilon }$ are the constants of HH and LH determined by material parameters including matrix elements for transition, exciton Rydberg energy, electron and hole effective masses and many fundamental constants [25]. Based on Equation (1), we can obtain the extinction coefficient κ which is related the absorption coefficient α by $\kappa (E)=0.98\times 10^{-5}\alpha (E)/E$.

As we all know, the refractive index η and absorption coefficient *α* are not independent, which are related to each other via well known Kramers–Krönig (K-K) relation, which is given by:(2)$$\eta (E)=1+\frac{\hbar c}{\pi }℘\int_{0}^{\infty }\frac{\alpha (E)}{E^{\prime }{}^{2}-E^{2}}dE^{\prime },$$
where *c* is the speed of light in vacuum. The coefficient of the integral equals 6.28 × 10^−6^ if α is given in units of cm^-1^ and *E* is expressed in eV. ℘ indicates the principal value of the integral.

In general, the reflectivity according to the complex refractive index η+iκ can be written as
(3)$$R(\eta ,\kappa ,E)=\frac{{\left[\eta (E)-1\right]}^{2}+\kappa^{2}(E)}{{\left[\eta (E)+1\right]}^{2}+\kappa^{2}(E)}.$$

According to Equations (1)–(3), we can calculate the reflectivity R(η,κ,E) before considering BF and BGR effects as a function of photon energy *E*.

Considering BF and BGR effects, the absorption coefficient $\alpha^{\prime }$ is written as
(4)$$\alpha^{\prime }(N,P,E)=\sum_{\nu }C_{\nu }[f_{\nu }(E_{V})-f_{C}(E_{C})]\sqrt{E-E_{g}^{\prime }/E},$$
where *N* and *P* are the electron and hole density excited by laser pulse. $f_{v}(E_{V})$ and $f_{c}(E_{C})$ are the hole and electron distribution functions, respectively, which are given by the Fermi–Dirac distribution functions. The factor $\left[f_{v}(E_{V})-f_{c}(E_{C})\right]$ represents the contribution of the BF effect. Note that the band gap energy after shrinkage is modified as $E_{g}^{\prime }=E_{g}-\Delta E_{g}$ due to the BGR effect. The value of the band gap shrinkage $\Delta E_{g}$ with the carrier density *N* is given as the improved form proposed by Wolff [26]. Based on Equations (2) and (4), we can obtain the refractive index $\eta^{\prime }$ and extinction coefficient $\kappa^{\prime }$ after considering BF and BGR effects. The reflectivity $R^{\prime }(\eta^{\prime },\kappa^{\prime },E)$ including BF and BGR effects can be calculated according to Equation (3) by virtue of $\kappa^{\prime }$ and $\eta^{\prime }$.

Considering the effects mentioned above, the change in the reflectivity ∆*R* due to the change in the complex the refractive index is followed by
(5)$$\Delta R=R^{\prime }(\eta^{\prime },\kappa^{\prime },E)-R(\eta ,\kappa ,E).$$

The calculated ratio of ΔR and R in GaAs based on Equations (1)–(5) with the carrier density $N=P=1\times 10^{17}/{\mathrm{cm}}^{3}$ was shown in Figure 2.

The absorption coefficient photo-excited by circular polarization laser pulse can be modified by adding a factor $S_{C}^{\pm }$ that is originated from electron spin, namely,
(6)$$\alpha^{\pm }(E)=\sum_{C}S_{C}{}^{\pm }\alpha (E).$$

Here, the superscript indicates photo-excitation by left (−) or right (+) circular polarization pulse. The subscript *C* represents electron spin up (↑) or spin down (↓). According to selection rules shown in Figure 1, S↑+=S↓−=14, S↓+=S↑−=34. Note that the hole spin is neglected in Equation (6) because it is much shorter than 1 ps. Spin dependence of the change of reflectivity $\Delta R^{\pm }=R^{\prime }{}^{\pm }(\eta^{\prime },\kappa^{\prime },E)-R^{\pm }(\eta ,\kappa ,E)$ will be obtained by modifying Equations (1)–(5) on the basis of Equation (6). The calculated results in GaAs with the carrier density $P=2\times 10^{17}/{\mathrm{cm}}^{3}$, $N^{+}=1.5\times 10^{17}/{\mathrm{cm}}^{3}$ and $N^{-}=0.5\times 10^{17}/{\mathrm{cm}}^{3}$, which was shown in Figure 4a.

## 4. Results and Discussion

The calculated Δ*R*/*R* excited by linear polarization pulse in GaAs crystal is shown in Figure 2. *E_g_* and *E_g_*′ labeled in Figure 2a are original and renormalized band gap energy and band gap shrinks $\Delta E_{g}=E_{g}^{\prime }-E_{g}$ about 10 meV with carrier density of 1 × 10^17^/cm^3^. The blue and red solid lines in the figure are from the contribution of BF effect and BGR effects, respectively, while the black solid line is the total signal. One can see from Figure 2a: (1) the PPR signal near the band gap rapidly changes due to the band edge effect; (2) the contribution from BGR effect is negative with the excitation photon energy greater than the original band gap energy, while the contributions from the BF effect is more complex: there are two critical values labeled E_C1_ and E_C2_ where the PPR changes between positive (PB) to negative (PA) signal. (3) The total value shows a large positive peak which is attributed to BGR effect at *E_g_*′, while a negative peak owing to BF effect at *E_g_*. (4) As shown in the inset of Figure 2a, which is the enlargement of the total signal near band edge, Δ*R*/*R* is a very sensitive function of photon energy. The magnitude and signature of the PPR signal is determined by the competition of BF and BGR effects. Figure 2b shows the calculated PPR signal in bulk GaAs with photo-excited carrier density of 1 × 10^17^/cm^3^ (red) and 2 × 10^17^/cm^3^ (black). One can see that the PPR signal is also sensitive to carrier density. According to the inset of Figure 2b, the critical value E_1_ and E_2_ at high photon energy where PPR signal changes from positive to negative is quite different, which becomes larger under higher carrier density. Note that the PPR signal excited by special photon energy between E_1_ and E_2_ will experience from absorption to bleaching with increasing the carrier density from 1 × 10^17^/cm^3^ to 2 × 10^17^/cm^3^.

Figure 3 shows the representative time dependence of experimental PPR signal in bulk GaAs (a) and CdTe (b) crystal with different photon energies, which gives how the PPR signal evolves with delay time and photon energy. According to Figure 3, one can see that the Δ*R* signal is sensitive to photon energy around the band gap edge. From Figure 3a, when excitation energy is larger than band gap energy (e.g., with 1.425 eV), the PPR signal is positive. On the other hand, photo-excited with photon energy very near band gap (e.g., with 1.409 eV and 1.404 eV), the PPR signal changes from positive to negative. The positive reflectivity kinetics (1.425 eV in GaAs) followed by two characteristic recovery processes: the faster one with time constant of several picoseconds, which is from the cooling of the hot carrier distribution and the slower one about a nanosecond, owing to electron hole recombination via various combination processes. In Figure 3b, the PPR signal in CdTe with photon energy larger than *E_g_* (1.512 eV) has the same tendency as that in GaAs. The red line shows that the PPR curve slowly rises, which is photo-excited by photon energy of 1.485 eV slightly smaller than *E_g_*. We inferred that the long rising time indicated that the defect state below band gap was filled. There is little PPR signal with photon energy of 1.473 eV, which means that 1.473 eV is a critical value where the signal experiences sign change.

In order to completely understand the carrier relaxation mechanisms, photon energy dependent on PPR was studied systematically. Many experimental PPR measurements are carried out with different photon energy with fixed carrier density at 1.0 × 10^17^/cm^3^ for GaAs and CdTe crystal, respectively. Figure 3c summarizes the extremum values of transient reflectivity around zero delay time as a function of incident pump photon energy. Photon energy dependence of ∆R in both GaAs (red) and CdTe (black) show the same tendency. With the photon energy decreasing in the measurement range, ∆*R* increases at first and then decreases after reaching a maximum value, at last becoming negative when photon energy is smaller than band gap *E_g_*. At higher photon energy, the signal also gradually decreases, so we can infer that the reflectivity signal will turn into a negative value if carrier density is high enough as predicted by our theory. However, we can’t reach the sign change owing to the limitation of the laser energy. The calculated data in GaAs was also plotted in Figure 3c (blue solid line) in order to compare with the experimental results. The experimental investigation and theoretical calculation show the same variation trend near the band gap of GaAs. The deviation of detailed photon energy possibly resulted from the absence of other effects, such as free carrier absorption.

In addition, the PPR signal also depends on carrier spin polarization based on Equation (6). As shown in Figure 4a, (σ^+^, σ^+^) and (σ^+^, σ^-^) were calculated from co- and counter-circular polarization PPR, which indicates the electron spin relaxation process of majority (spin down) and minority (spin up), respectively. Both (σ^+^, σ^+^) and (σ^+^, σ^-^) curves show similar trend and different magnitude. The critical values labeled E_1_ and E_2_ in the inset where $\Delta R^{\pm }/R^{\pm }$ changes between positive to negative signal was different. Time resolved circularly polarized PPR is one of the powerful methods to investigate carrier spin lifetime and a relaxation mechanism. Spin dependent PPR signal in CdTe was shown in Figure 4b with carrier density of 2.0 × 10^17^/cm^3^. The photon energy is 1.6 eV. The black (σ^+^, σ^+^) and red (σ^+^, σ^-^) hollow (○) lines are obtained by co- and counter-circular polarization pump probe, respectively. (σ^+^, σ^+^) curve shows that PPR is photo-bleaching and (σ^+^, σ^-^) shows that PPR is photo-absorption. We can infer the photon energy used here just between E_1_ and E_2_ shown in Figure 4a according to the experimental results. This phenomenon was also observed in InP single crystal with carrier density of 1 × 10^17^/cm^3^ at low temperature (70K) [17]. We globally fitted the difference of (σ^+^, σ^+^) and (σ^+^, σ^-^) (blue □ line) by a single exponential and obtained the electron lifetime t_s_ of ~3 ps, which is the same as the previous report in CdTe [13] and faster than that in bulk GaAs [23] and InP [17]. The spin life time is one of the important parameters for designing spintronics devices. Such a short spin life meets the demands for ultrafast spin optical switching.

## 5. Conclusions

This work systematically investigated the spin dependent PPR by taking BF and BGR effects into account in intrinsic GaAs crystal at room temperature. Assuming parabolic absorption band structure, the refractive index was obtained by virtue of the K-K relation. We discussed the separate contributions of BF and BGR effect for linear polarization PPR. The most interesting property is that there is a critical photon energy where the signal experiences sign change between bleaching and absorption owing to the competition of BF and BGR effects in the total PPR signal of GaAs. Time resolved pump probe reflectivity spectroscopy near the band edge was conducted in GaAs and CdTe crystals with 100 fs laser pulse. The extremum values of PPR curves with different photon energies were extracted as a function of photon energy, which has the similar trend as our model. In addition, the PPR signal is sensitive to electron spin orientation. The PPR signal from co- and counter-circularly polarized light is completely different, which agrees well with our experimental data from circular polarization PPR. Our research in this work provides a method to study optoelectronic properties of conventional semiconductors under a moderate carrier density. Importantly, this model can be used to other novel semiconductor materials beyond GaAs and will open up more avenues of research that will provide new insights into the underlying photophysics.

## Figures and Tables

**Figure 1 materials-13-00242-f001:**
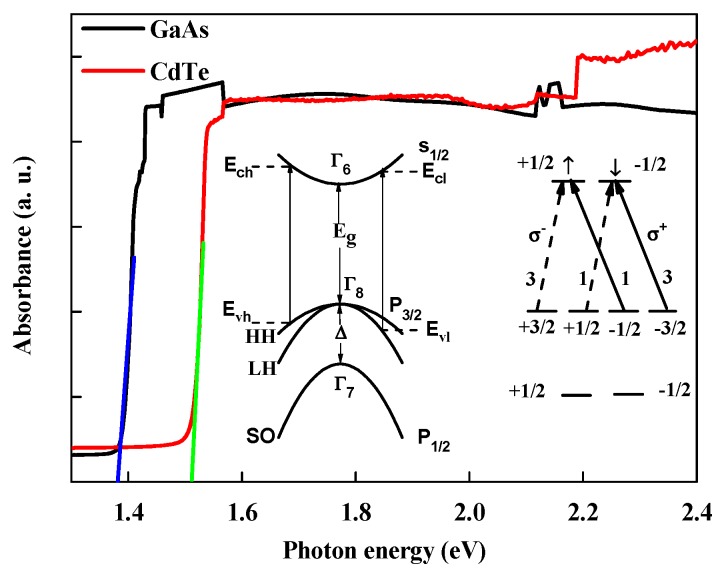
Absorption spectra in bulk GaAs (black line) and CdTe (red line) single crystal. Estimation band gap energies (~1.51 eV and 1.38 eV for CdTe and GaAs) from the Tauc plot. The insets are band structure at Γ point of the first Brillouin zone, selection rules and relative intensities for right-handed $\sigma^{+}$ (solid arrow) and left-handed $\sigma^{-}$ (dashed arrow) circularly polarized light.

**Figure 2 materials-13-00242-f002:**
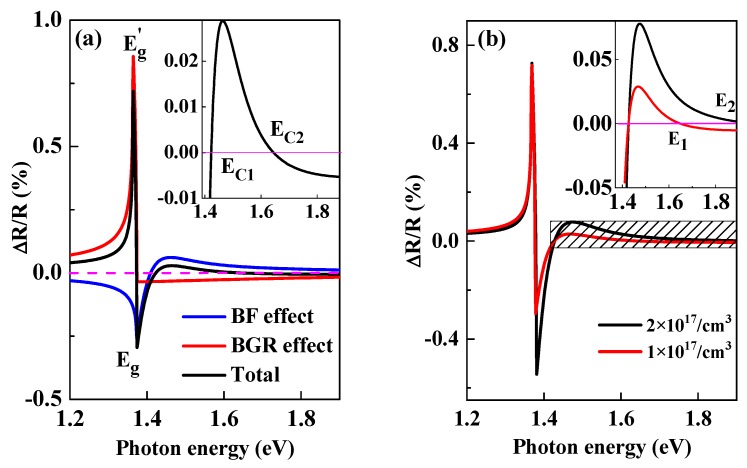
(**a**) the calculated reflectivity change in GaAs with carrier densities of 1 × 10^17^/cm^3^ (black line). The blue and red lines are calculated from BF and BGR effects, respectively; (**b**) the reflectivity change with the carrier density of 1 × 10^17^/cm^3^ (red) and 2 × 10^17^/cm^3^ (black). The inset is the enlargement of the shading area. The pink line at zero was a guide for the eyes.

**Figure 3 materials-13-00242-f003:**
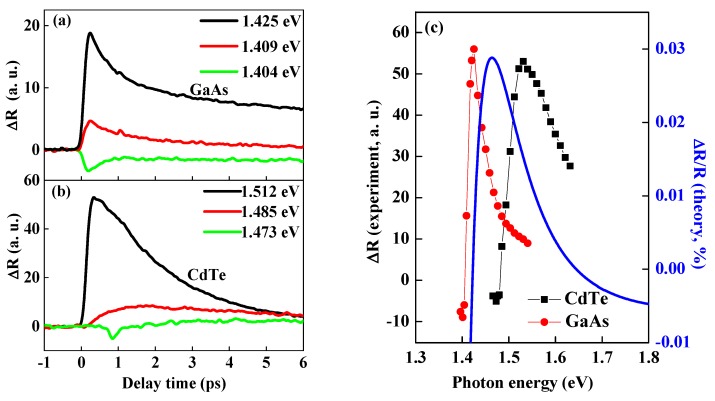
The representative time resolved pump probe reflectivity spectroscopy near band gap energy in GaAs crystal (**a**) and CdTe crystal (**b**); (**c**) photon energy dependence of reflectivity spectroscopy of CdTe (black ■) and GaAs (red •) crystal. The blue solid line is the calculated results based on Equation (5) in GaAs for comparison with the experimental values.

**Figure 4 materials-13-00242-f004:**
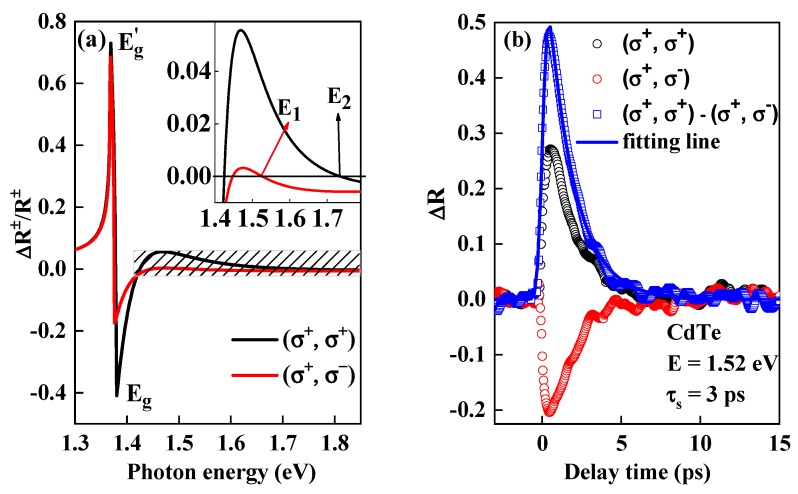
(**a**) The black and red lines are calculated from co- and counter-circularly polarized pump and probe, respectively. The hole density is set to be 2 × 10^17^/cm^3^, the electron density from HH and LH are 1.5 × 10^17^/cm^3^ and 0.5 × 10^17^/cm^3^, respectively. The inset is the enlargement of the shading area. E_1_ and E_2_ are the critical photon energy; (**b**) typical co- (black ○, (σ^+^, σ^+^)) and counter- (red ○, (σ^+^, σ^-^)) circular polarization pump probe in CdTe. The blue (□) line is the difference of (σ^+^, σ^+^) and (σ^+^, σ^-^), which showed electron spin relaxation with delay time. The blue solid line is globally fitted by a single exponential function considering the system response time. The fitted lifetime is ~3 ps.

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
