# Peer review of "Femtosecond Pump Probe Reflectivity Spectra in CdTe and GaAs Crystals at Room Temperature"

_materials, 2020, doi:10.3390/ma13010242_

Round 1
Reviewer 1 Report
The paper could be of interest for Materials, but the following revisions have to be considered before acceptance:
-English should be extensively revised. Full sentences appear meaningless andshould be rewritten (such as, in the introduction: “Also, there is something interesting is only partially confirmed in experiment up to now, which has a lot of space to be verified further. ”).
-What is the meaning of the pink lines in Figure 2?
Author Response
Q1:English should be extensively revised. Full sentences appear meaningless and should be rewritten (such as, in the introduction: “Also, there is something interesting is only partially confirmed in experiment up to now, which has a lot of space to be verified further. ”).
A: Thanks to the reviewer for such a honest suggestion. We have checked the manuscript thoroughly and tried our best to improved English presentation in the revised manuscript.
Q2: What is the meaning of the pink lines in Figure 2?
A: The pink lines at zero was guide for eyes.
Reviewer 2 Report
The paper gives a cursory description of some pump-probe reflectivity measurements. The experiment itself is reasonably standard, and appears to have been well-performed. The data analysis and theoretical modeling is also reasonably standard. Unfortunately, it is not clear what scientific question was being asked or what was learned; the paper feels as though they took some measurements for no particular reason and wrote up what they saw. The problem is particularly evident in the conclusions section – the authors have no conclusions at all! Besides the overall lack of a scientific motivation, the paper has several specific problems that could be reasonably easily fixed; these are described below.
In section 2 they show absorption spectra of their samples, but there are odd features in the data that are unexplained, particularly in GaAs. Are these real, or is the optical density so high that these are instrumental artifacts? The reader cannot tell, because the vertical axis is given in arbitrary units, for no obvious reason. If the authors are going to show this data, the vertical axis should be in real units and strange features should be explained; otherwise, do not show this data.
Section 3 gives a reasonably succinct description of the theoretical model. There is one major problem that should be fixed, however. In figure 2a the authors separately show a BGR effect and a BF effect, but they give no indication in their theory section how they separately calculate such things. If the authors are going to show all these equations, they should show the ones that actually contribute to their displayed calculations. The other problem is that the calculation and experiment shown in figure 3c do not match. In fact, they are not even close, and cannot be explained away simply by stating that it may be due to free carrier absorption (as suggested on line 210).
The data in figure 3b is quite interesting, and is completely unexplained in the text. The closest the authors come is line 193, when it is falsely claimed that CdTe behaves somewhat similar to GaAs. Why is there any signal at all at 1.485eV and 1.473eV, which are below the bandgap of the system? The red line is reasonably interesting – it shows no signal at all at time zero, followed by a slow rise. This clearly is some effect completely unaccounted for by the theoretical model in section 3, which assumes instantaneous creation and dephasing of carriers. The green line shows what looks like some instrumental artifact of some kind? If the signal in the green line is real, it too is completely unaccounted for by the proposed theoretical model, and the paper is completely silent on this topic.
On figure 3c, what is the time delay? The text makes it sound as though data are taken at different time delays for different photon energies, but it is not clear on this point. Transient spectra are only meaningful if the points are all taken from the same time delay, and are only meaningful if that time delay is given.
In figure 4b, there should be some mention either in the legend or in the caption of what the blue squares represent. I should not have had to read the text to find out it was the black trace minus the red trace.
Round 2
Reviewer 1 Report
I suggest that the paper is published